# AQoLS-Brief: Development and psychometric properties of a short version of the alcohol quality of life scale

**Amandine Luquiens** [1,2*], **Henri Panjo** [2], **Alexandra Dereux** [3], **Patrice Louville** [4], **Hélène Donnadieu** [5], **Marie Bronnec** [6], **Amine Benyamina** [7], **Pascal Perney** [1], **Arnaud Carré** [8], **Nathalie Pelletier-Fleury** [2]

**1** University of Montpellier. Department of Addiction, Universitary Hospital of Nîmes, Nîmes, France, **2** CESP, University Paris-Sud, UVSQ, INSERM, Université Paris-Saclay, Villejuif, France, **3** Assistance publique -Hôpitaux de Paris, GHU.NORD, Hôpital Fernand Widal, Département de Psychiatrie et de Médecine Addictologique, Paris, INSERM UMRS, Université Paris Cité, Paris, France, **4** Service de Psychiatrie et d'Addictologie de l'Adulte et du Sujet Âgé, hôpital Corentin-Celton, G.H.U. AP-HP. Centre – Université Paris Cité, Issy-les-Moulineaux, France, **5** Department of Addiction, Universitary Hospital of Montpellier, University of Montpellier. Pathogenesis and Control of Chronic & Emerging Infections, University of Montpellier, INSERM, Montpellier, France, **6** Nantes Université, University Hospital of Nantes, UIC Psychiatrie et Santé Mentale, Nantes, France, Nantes Université, Univ Tours, Universitary Hospital of Nantes, INSERM, MethodS in Patient-centered outcomes and HEalth ResEarch, SPHERE, Nantes, France, **7** Department of Psychiatry and Addictology, Paul Brousse Hospital, UR PSYCOMADD University Paris-SUD, Villejuif, France, **8** University Savoie Mont Blanc, Univ. Grenoble Alpes, LIP/PC2S, Grenoble, France

* amandineluquiens@gmail.com

## Abstract

### Background

Short versions of health-related quality of life measures are essential for the development of preference-based measures used to obtain the quality-adjustment weights needed to calculate quality-adjusted life years (QALYs) in health economic evaluations. Using data from the randomized TRAIN study, which compared the efficacy of two cognitive training programs in patients with alcohol use disorder (AUD) recently detoxified at five centers in France from 2019 to 2023 (n=227), we report the development and psychometrics of the AQoLS-Brief, a short version of the initial 34-item Alcohol Quality of Life Scale (AQoLS).

### Methods

Baseline data on, age, sex, AUD severity, drinking characteristics and consequences of alcohol including the AQoLS were collected. One item per dimension of the AQoLS was chosen following predefined rules. Construct validity of the AQoLS-brief was documented: internal consistency was assessed using Cronbach's alpha and external validity was assessed with external drinking and non-drinking variables via univariate analyses, a confirmatory factorial analysis (CFA) and a multivariate linear regression.

### Results:

The 7-item AQoLS-Brief demonstrated good internal consistency (Cronbach alpha= 0.80) and very high correlation with the long version AQoLS (rho=0.95). The CFA confirmed

**Data availability statement:** Anonymized data can be shared upon reasonable request. In France, the sharing of clinical research data is governed by the General Data Protection Regulation (GDPR) and national regulations that prioritize the protection of personal and sensitive health information. Consequently, access to such data is only granted upon reasonable request, requiring a clear and justified purpose in line with the original research objectives. This framework ensures compliance with strict legal and ethical standards, safeguarding participants' confidentiality while allowing for responsible data sharing to advance scientific research. Data will be securely stored on a virtual server dedicated to the project, with access provided upon request to researchers through Frédéric Robergeau (frederic.robergeau@ inserm.fr), head of the IT department at CESP. Researchers will have remote access secured by two-factor authentication. However, they will not have the ability to extract or copy-paste the data. Upon request, tailored analysis packages (e.g., R packages) will be made available to them for their analyses. The results of these analyses will be transmitted directly by Frédéric Robergeau.

**Funding:** The TRAIN study was funded by a French national grant of the ministry of health PHRC-N 2016.

**Competing interests:** AL, HP, MB, AD, PP, AC, NPF: no conflict of interest related to this manuscript HD: declares to have participated in occasional interventions (conference activities) for Abbvie, Gilead, Ethypharm and consultancy for Ethypharm AB: declares to have participated in occasional interventions (conference activities) for Eisai, Gilead and Abbvie. Board member Indivior and Camurus PL: declares to have participated in occasional interventions (conference activities) for Lundbeck This does not alter our adherence to PLOS ONE policies on sharing data and materials.

**Abbreviations:** AQoLS-Brief: Alcohol Quality of Life Scale; AUD: alcohol use disorder; CGI-S: Clinical Global Impressions - Severity; PROM: patient reported outcome measure; QALY: Quality-Adjusted Life Year; ANPS: Affective Neuroscience Personality Scales; SURPS: Substance Use Risk Profile Scale.

the unidimensionality of the AQoLS-Brief. Severity of AUD, as measured by the number of DSM-5 alcohol criteria and the number of heavy drinking days, and the dimension Sadness of the Affective Neuroscience Personality Scales were the heaviest contributors to the AQoLS-Brief score.

## Conclusions:

The AQoLS-Brief is a short version of the AQoLS with good psychometrics, enhancing its use in clinical practice and research, and representing a significant step toward establishing a relevant and specific preference-based measures for calculating QALYs in AUD populations.

## Background

The Alcohol Quality of Life Scale (AQoLS) is a patient-reported outcome measure (PROM) developed with significant input from patients with current or remitted alcohol use disorder (AUD)[1], in accordance with COSMIN (COnsensus-based Standards for the selection of health Measurement INstruments) guidelines [2]. These guidelines emphasize ensuring strong content validity by involving patients in identifying relevant elements through concept elicitation and conducting cognitive interviews to assess the comprehensiveness, comprehensibility, and relevance of the items in alignment with the intended construct.

The 7-dimension structure comprising 34 items was described in an exploratory analysis in a population of French patients with current or remitted AUD (n=285): "activities", "relationships", "living conditions", "negative emotions", "self-esteem", "control" and "sleep" [3]. Since its development and validation, the AQoLS has been used in several observational studies in the general population [4], patient populations [5–7], and clinical trials [8] and is the most frequently used alcohol-specific health-related quality of life measure [9]. It is available and validated in several languages (French, English, Japanese, Chinese, Korean, Spanish, Italian) [4,10]. The AQoLS has shown excellent internal consistency [11]. The AQoLS total score showed an intraclass correlation coefficient at 2 weeks in stable out-patients with alcohol use disorder of 0.80 [12]. However, its length could limit its use in clinical trials, where subjective measurements are not often the primary judgment outcome [13] and numerous time-consuming measures are typically involved. A shorter version would be more convenient for both researchers and participants. Using a shorter version of a scale is more convenient as it reduces the time burden on both participants and researchers, making it easier to administer in clinical or research settings where time constraints are often present. A shorter scale also improves participant engagement and completion rates, while still retaining the key measurement properties and relevance of the longer version. This helps ensure more efficient data collection without compromising the quality of the information gathered.

Short versions of health-related quality of life measures can also be useful to develop preference-based measures. Preference-based measures are used to obtain the quality-adjustment weights needed to calculate quality-adjusted life years (QALYs) in health economic evaluations [14]. Preference-based measures typically consist of a self-administered patient questionnaire, a health state classification system, and preference weights for all states defined by the classification system [15] and often developed from generic health-related quality of life measures such as Short Form 36-Item Health Survey (SF-36) which gave rise to SF6D, or the EuroQol 5-Dimension (EQ-5D) [16]. However, generic instruments may lack the coverage to detect changes in important aspects of certain conditions, for instance cognition

or relationships with family and social support [13,17]. The AQoLS showed moderate-to-low correlation with the EQ-5D and the SF-36 [3].

Utility values for various health states reflecting the spectrum of AUD, derived from the AQoLS, could be incorporated into decision-analytical models of AUD through the estimation of QALYs. However, developing such a data set from a long-form scale poses methodological challenges, particularly in designing preference tasks for discrete choice experiments, a method increasingly used to derive utilities. Specifically, each item of a preference-based measure becomes an attribute in a discrete choice experiment. Short versions of scales are needed for a discrete choice experiment due to the complexity of constructing pairwise combinations. These combinations involve comparing multiple health states defined by different attributes, and as the number of attributes and levels increases, the total number of possible pairwise combinations grows exponentially. By using condensed versions of scales, which reduces the number of attributes and levels, the number of combinations is significantly reduced, making the experimental design more manageable. This simplification helps avoid participant fatigue and ensures the choice tasks are feasible and understandable, leading to more reliable data for deriving utility values. A recent systematic review of discrete choice experiments related to health technologies found that the average number of attributes used was approximately six [18].

We therefore aim to develop a short version of the AQoLS, the AQoLS-Brief, as a concise and practical measure of quality of life in individuals with AUD, and as a first step for a future specific health utility score. We report here the development and psychometrics of the AQoLS-Brief which we carried out using data from the Train study (awaiting publication, ongoing statistical analyses).

## Methods

### Study design

This study is an *ad hoc* analysis of baseline data from the TRAIN study (TRAining INhibition in alcohol use disorder). The TRAIN study is a randomized clinical trial comparing the efficacy in patients with AUD recently detoxified, of a computerized cognitive training program targeting executive control versus a sham program delivered twice a week, over six weeks, with weekly debriefings by a neuropsychologist. The main outcomes of the study will be published at a later date.

### Compliance with ethical standards

All procedures were in accordance with the ethical standards of the responsible committee on human experimentation (institutional and national) and with the Helsinki Declaration of 1975, as revised in 2000. Written informed consent was obtained from all patients included in the study. The clinicaltrials.gov registration number is NCT03530384. The TRAIN study was approved with the N° IDRCB: CPP18-013b/2017-A03558-45 on the 30/05/2018 by the French ethics committee "Comité de protection des Personnes".

### Population

We included all patients from the TRAIN study, recruited in five out- and in-patient centers specialized in addiction in France from 1 February 2019 to 12 January 2023, who had completed the AQoLS at baseline (n=227).

Patient had to be aged 18–65 with current AUD, according to the DSM-5. They had to be classed at least as high drinking risk level (men: alcohol consumption >60 g/day; women

>40 g/day) as regarding the alcohol consumption in the last 4-week drinking period. Patient had to be recently detoxified and abstinent since 7–30 days at inclusion, regardless of their drinking goal for the following period. Inclusion criteria also included no benzodiazepines use for at least 3 days prior to inclusion, to avoid interference with alcohol intoxication or withdrawal medication on the neuropsychological assessments, and any recall bias on drinking.

Exclusion criteria were current alcohol withdrawal symptoms (Cushman score > 3)[19], other addictions (excluding tobacco), and psychiatric comorbidity (psychotic disorders, current manic/hypomanic episode, current major depressive episode), as assessed with the MINI, Alzheimer disease, Korsakoff syndrome, mental retardation, or any condition that may significantly alter the computerized-task completion, as assessed by the clinician. Patients were also excluded if they were unable or unwilling to comply with the protocol requirements.

## Measures

### Sociodemographic variables (age, sex) were collected by the clinician

**History of the alcohol use disorder.** The time since the onset of the alcohol use disorder according to the patients was collected (years).

**Consequences of alcohol.** The AQoLS [1,11] is a 34-item patient-reported outcome measuring health-related quality of life, specific to patients with AUD, developed from the patients' perspective, ranging from 0 (lowest) to 102 (highest subjective impact). The AQoLS has seven dimensions comprising the following items:

- Activities: Q2. Plans, Q3. Restricted in places, Q4. Physical activities, Q5. Jobs around the house, Q6. Ability to work, Q7. Cut myself off, Q13. Sex, Q15. Household affairs, Q25. Appearance, Q26. General health

- Relationships: Q1. Everyday activities, Q8. Neglected people, Q9. Relationships, Q10. Behaved badly, Q11. Family, Q27. Risky situations

- Living conditions: Q16. Housing situation, Q17. Money spent on alcohol, Q18. Financial difficulties, Q24. Appetite

- Negative emotions: Q22. Worried about my health, Q23. Worried about my life

- Self-esteem: Q12. Trust, Q14. Friends, Q19. Shame, Q20. Contempt, Q21. Wasting my life

- Control: Q28. Nothing matters, Q29. Alcohol controlled me, Q30. Life around alcohol, Q31. Plan around alcohol, Q32. Control of myself

- Sleep: Q33. Good night's sleep, Q34. Sleep enough

Each item is scored on a 4-point Likert scale (0–3), with a recall period of last 4 weeks, and a detailed instruction asking completers to focus on their relation to alcohol to answer, whatever their drinking status in the last 4 weeks.

The Montreal Cognitive Assessment (MoCA) includes 13 tasks measuring the following 8 cognitive domains: visuospatial skills/executive function, naming, immediate memory, attention, language, abstraction, delayed recall, and orientation. A total score is calculated by summing the scores of the 13 tasks. The maximum score possible is 30 points. The normal value in AUD patients is ≥ 26 and the score does not need to be corrected in those with a low education level (Ewert et al., 2018).

**Severity of the alcohol use disorder.** The number of DSM-5 criteria for AUD (maximum=11; minimum for diagnosis=2) [20] and the Clinical Global Impressions-Severity (CGI-S) scale were used to assess disease severity from the perspective of the clinician

[21]. The CGI-S was assessed for the current state by the clinician on a 7-point Likert scale (minimum 1 (normal, i.e., no symptoms since 7 days); maximum 7 (i.e., amongst the most severely ill)).

**Drinking outcomes.** We collected data on the time elapsed since the last drink and on alcohol consumption during the so-called 4-week drinking period leading up to the last drinking occasion including the number of heavy drinking days (HDD), i.e., 40 g or more if a female, 50 g or more if a male, with the Time Line Follow Back method [22].

**Psychological variables.** We collected the Affective Neuroscience Personality Scales (ANPS) and the Substance Use Risk Profile Scale (SURPS) to assess personality traits. The ANPS measures six primary emotional traits: seeking, care, play, fear, anger, and sadness. It was initially validated by Davis and Panksepp [23] providing a framework to explore the neurobiological basis of personality. The SURPS assesses four personality dimensions associated with substance use vulnerability: anxiety sensitivity, hopelessness, impulsivity, and sensation seeking, with its initial validation conducted by Woicik et al. [24]. Both scales are psychometrically robust and widely used in research to link personality traits to behavioral and emotional outcomes.

## Statistical analyses

**Selection of the items for the short version.** We selected one item per dimension using *a priori* rules demonstrating the predominant nature of the item within each dimension. This choice was made to respect the input of patients at the time of the development of the AQoLS, that lead in a qualitative analysis to the identification of the 7 dimensions [1] confirmed in the later validation [11]. For each dimension, we chose the item best combining the following rules:

1. Cronbach α of the dimension without the item is the lowest

2. Item-rest correlation (ir-ρ), i.e., correlation between the item and the dimension's score formed by all other items, is the highest. ir-ρ is an index of item discrimination.

For items with tied score for either rule, the choice was based on the second rule [25] and clinical relevance (AL and HP) to choose the most universal item, i.e., the selected item should be the less sensitive to lifestyle, marital status or professional situation.

To further document the relevance of the selection, we calculated additional psychometrics on the 184 item difficulty and discrimination. Item difficulty was assessed by item distribution and the average 185 item score. In order to obtain item difficulty indices (noted $d$) comparable across all items in a multiple-item instrument we scaled each average item score to interval [0,1] by accounting for a minimal and maximal possible item score min (0) and max (3) [26]. The item difficulty should lie between 0.5 and 0.8. Item discrimination was estimated by the upper-lower index, the difference in difficulty between an *upper* and *lower* group of respondents based upon their respective dimension scores [26]. By a rule of thumb, upper-lower index based upon thirds should not be lower than 0.2, except for very easy or very difficult items [27]. Items of medium difficulty with lower upper-lower index are considered suspicious, and their content and wording should be checked.

We also generated all possible combinations of the seven items with one item per dimension, i.e., 24000 possible questionnaires, and retained the versions with the maximum possible Cronbach α to compare against the version obtained with the *a priori* rules.

**Internal consistency and external validity.** We calculated the Cronbach α and the average inter-item correlation for the AQoLS-Brief, the AQoLS and each of its dimensions to further document the relevance of the item selection. Average inter-item correlation

for a set of items should be between 0.20 and 0.40, suggesting that while the items are reasonably homogenous, they do contain sufficiently unique variance so as to not be isomorphic with each other. For every item we also compute α-drop and average inter-item correlation--drop which are respectively define as Cronbach α excluding the particular item of the dimension and AIIC excluding the particular item of the dimension. The lowest the α-drop/ average inter-item correlation -drop for an item is, the more important this item is for this dimension. We also determined the Spearman correlation coefficient between AQoLS and AQoLS-Brief. Finally, we performed a confirmatory factor analysis (CFA) of the AQoLS-Brief to validate the unidimensionality of the construct being measured. We estimated the CFA model with the R lavaan package [28] treating the seven items of AQoLS-Brief as ordinal variables [29]). It uses the diagonally weighted least squares method for the estimation which is encouraged when the sample is small and data violates normality [30]. To assess the fit of the CFA model, the chi-square test, root mean square error of approximation, standardized root mean square residual, Tucker-Lewis Index, and Comparative Fit Index were examined. These measures were evaluated against established guidelines to determine the adequacy of the model fit.We explored the external validity calculating the Spearman correlations between AQoLS/AQoLS-Brief and external variables [31]. We hypothesized a moderate positive correlation with measures of severity (CGI-severity and DSM-5 criteria for AUD), a moderate to low positive correlation with the history of alcohol use disorder (i.e., the length since the onset of alcohol use disorder), the drinking outcome (i.e., the number of HDD in the last 4-week drinking period) and dimensions of the psychological scales ANPS and SURPS related to anxiety-depression, i.e., ANPS Sadness and Fear dimensions, and SURPS Anxiety Sensitivity and Hopelessness dimensions. We expected no correlation with the cognitive impairment as assessed with the MoCA score. We performed a multiple linear regression explaining the AQoLS-brief/ AQoLS by the external variables.

All analyses were performed using R Statistical Software (v4.4.2; R Core Team 2024).

## Compliance with Ethical Standards

All procedures followed were in accordance with the ethical standards of the responsible committee on human experimentation (institutional and national) and with the Helsinki Declaration of 1975, as revised in 2000. Informed consent was obtained from all patients for being included in the study. The clinicaltrials registration number is NCT03530384. The study was approved with the N° IDRCB: 2017-A03558-45 and SI N° 18.01333.201813-MS04 by the French ethic committee "Comité de protection des Personnes".

## Results

Table 1 presents the sample characteristics. Patients were predominantly male (66.1%), with an average age of 49.7 years. The average number of DSM-5 criteria fulfilled for alcohol was 8.6, and the CGI-severity score showed highest scores for normal (44%), followed by markedly ill (25%). All participants completed the questionnaire without missing items.

Table 2 presents AQoLS total score and dimensions subscores.

## Selection of the items for the short version

Table 3 presents the results of the selection of the items for the AQoLS-Brief following the *a priori* rules and additional psychometrics on item difficulty, discrimination and internal consistency.

Two dimensions showed ties on the *a priori* rule 1.

**Table 1. Sample characteristics.**

| Variable | N=227 |
|---|---|
| Age (years) (mean, sd) | 49.7 (9.1) |
| Gender (male) % (n) | 66.1% (150) |
| MoCA score (mean, sd) | 26.4 (2.7) |
| Low education level (less of 12 years) % (n) | 97 (43.1%) |
| ANPS total score (mean, sd) | 63.3 (9.7) |
| Sadness | 10.2 (3.3) |
| Fear | 10.1 (3.7) |
| Care | 12.8 (3.1) |
| Anger | 7.4 (3.5) |
| Play | 10.9 (3.3) |
| Seeking | 11.9 (3.1) |
| SURPS Total score (mean, sd) | 56.1 (8.0) |
| Hopelessness | 16.7 (4.3) |
| Anxiety Sensitivity | 15.2 (2.9) |
| Sensation Seeking | 13.0 (3.4) |
| Impulsivity | 11.4 (3.0) |
| Length since the onset of alcohol use disorder (years) (mean, sd) | 14.7 (11.0) |
| Number of days since last drink (mean, sd) | 17.5 (6.7) |
| Number of heavy drinking days (28d) (mean, sd) | 22.3 (8.7) |
| Number of DSM-5 alcohol criteria (mean, sd) | 8.6 (2.2) |
| CGI-severity % (n) | |
| Normal, no symptom since 7 days | 44.1% (100) |
| Almost not mentally ill | 4.0% (9) |
| Mildly ill | 4.4% (10) |
| Moderately ill | 10.1% (23) |
| Markedly ill | 24.7% (56) |
| Severely ill | 8.4% (19) |
| Among the most extremely ill patients | 0.9% (2) |
| Not evaluated | 3.5% (8) |

**Table 2. AQoLS scores.**

| Score | Mean [95% CI] |
|---|---|
| AQoLS-Total | 48.5 [45.8; 51.1] |
| Activities | 14.2 [13.3; 15.1] |
| Relationships | 8.5 [7.9; 9.0] |
| Living conditions | 3.9 [3.5; 4.3] |
| Negative emotions | 3.7 [3.5; 3.9] |
| Self-esteem | 7.5 [7.0; 7.9] |
| Control | 7.5 [7.0; 8.0] |
| Sleep | 3.2 [2.9; 3.5] |

The dimension "Relationships" was a tie between with item 9 "Alcohol has damaged my close relationships" and item 11 "I have felt I miss out on family life because of alcohol" *ex aequo* for rule 1. We kept item 11 because it did better on rule 2.

**Table 3.  Psychometrics of AQoLS items in each dimension and selected items in the AQoLS-Brief.**

| Item | ir-ρ[a] | Upper-lower index | Cronbach α[b] | Average inter-item correlation [c] | d[d] | Item distribution |
|---|---|---|---|---|---|---|
| **Activities** | | | 0.87 | 0.40 | | |
| 02. Plans | 0.58 [0.49; 0.66] | 0.46 | 0.86 | 0.40 | 0.48 | 15.4 \| 37.9 \| 34.4 \| 12.3 |
| 03. Restricted in places | 0.52 [0.42; 0.61] | 0.51 | 0.86 | 0.41 | 0.42 | 31.3 \| 29.5 \| 22.0 \| 17.2 |
| **04. Physical activities** | **0.74 [0.67; 0.79]** | **0.67** | **0.84** | **0.38** | **0.51** | **21.6 \| 27.3 \| 26.4 \| 24.7** |
| 05. Jobs around the house | 0.67 [0.59; 0.74] | 0.64 | 0.85 | 0.39 | 0.46 | 24.2 \| 34.4 \| 21.1 \| 20.3 |
| 06. Ability to work | 0.60 [0.51; 0.68] | 0.53 | 0.86 | 0.40 | 0.46 | 22.5 \| 31.7 \| 30.0 \| 15.9 |
| 07. Cut myself off | 0.60 [0.51; 0.68] | 0.55 | 0.86 | 0.40 | 0.54 | 16.3 \| 30.4 \| 28.2 \| 25.1 |
| 13. Sex | 0.38 [0.26; 0.49] | 0.39 | 0.87 | 0.44 | 0.42 | 29.1 \| 29.5 \| 26.4 \| 15.0 |
| 15. Household affairs | 0.60 [0.51; 0.68] | 0.59 | 0.86 | 0.40 | 0.48 | 22.9 \| 33.0 \| 21.1 \| 22.9 |
| 25. Appearance | 0.59 [0.50; 0.67] | 0.52 | 0.86 | 0.40 | 0.40 | 30.0 \| 32.6 \| 24.2 \| 13.2 |
| 26. General health | 0.62 [0.53; 0.69] | 0.47 | 0.86 | 0.40 | 0.54 | 11.5 \| 30.8 \| 40.5 \| 17.2 |
| **Relationships** | | | 0.83 | 0.45 | | |
| 01. Everyday activities | 0.51 [0.41; 0.60] | 0.48 | 0.82 | 0.48 | 0.58 | 09.7 \| 30.4 \| 35.7 \| 24.2 |
| 08. Neglected people | 0.63 [0.54; 0.70] | 0.51 | 0.79 | 0.43 | 0.43 | 23.3 \| 35.7 \| 29.1 \| 11.9 |
| 09. Relationships | 0.69 [0.62; 0.75] | 0.63 | 0.78 | 0.42 | 0.49 | 18.9 \| 34.8 \| 26.9 \| 19.4 |
| 10. Behaved badly | 0.60 [0.51; 0.68] | 0.47 | 0.80 | 0.44 | 0.37 | 27.3 \| 41.9 \| 22.0 \| 08.8 |
| **11. Family** | **0.72 [0.65; 0.78]** | **0.63** | **0.78** | **0.41** | **0.53** | **16.7 \| 31.7 \| 28.2 \| 23.3** |
| 27. Risky situations | 0.45 [0.34; 0.55] | 0.44 | 0.83 | 0.49 | 0.42 | 24.7 \| 36.6 \| 27.3 \| 11.5 |
| **Living conditions** | | | 0.75 | 0.43 | | |
| 16. Housing situation | 0.46 [0.35; 0.56] | 0.43 | 0.74 | 0.49 | 0.21 | 61.2 \| 22.5 \| 07.5 \| 08.8 |
| 17. Money spent on alcohol | 0.65 [0.57; 0.72] | 0.61 | 0.63 | 0.37 | 0.36 | 36.1 \| 24.7 \| 34.8 \| 04.4 |
| **18. Financial difficulties** | **0.67 [0.59; 0.74]** | **0.56** | **0.62** | **0.36** | **0.30** | **44.9 \| 27.8 \| 20.7 \| 06.6** |
| 24. Appetite | 0.42 [0.31; 0.52] | 0.55 | 0.76 | 0.51 | 0.44 | 26.4 \| 28.6 \| 30.8 \| 14.1 |
| **Negative emotions** | | | 0.69 | 0.53 | | |
| 22. Worried about my health | 0.53 [0.43; 0.62] | 0.56 | 0.60 | | 0.59 | 08.4 \| 32.6 \| 32.2 \| 26.9 |
| **23. Worried about my life** | **0.53 [0.43; 0.62]** | **0.51** | **0.47** | | **0.64** | **04.4 \| 24.7 \| 44.5 \| 26.4** |
| **Self-esteem** | | | 0.81 | 0.45 | | |
| 12. Trust | 0.59 [0.50; 0.67] | 0.55 | 0.77 | 0.46 | 0.43 | 24.2 \| 36.6 \| 25.1 \| 14.1 |
| 14. Friends | 0.56 [0.46; 0.64] | 0.49 | 0.78 | 0.47 | 0.37 | 27.8 \| 40.5 \| 23.3 \| 08.4 |
| 19. Shame | 0.60 [0.51; 0.68] | 0.53 | 0.76 | 0.45 | 0.70 | 04.0 \| 24.2 \| 30.8 \| 41.0 |
| 20. Contempt | 0.56 [0.46; 0.64] | 0.45 | 0.78 | 0.47 | 0.34 | 32.2 \| 41.4 \| 19.8 \| 06.6 |
| **21. Wasting my life** | **0.64 [0.56; 0.71]** | **0.55** | **0.75** | **0.43** | **0.65** | **05.7 \| 25.6 \| 36.6 \| 32.2** |
| **Control** | | | 0.86 | 0.55 | | |
| 28. Nothing matters | 0.70 [0.63; 0.76] | 0.60 | 0.82 | 0.54 | 0.42 | 26.4 \| 32.2 \| 30.0 \| 11.5 |
| 29. Alcohol controlled me | 0.71 [0.64; 0.77] | 0.60 | 0.82 | 0.54 | 0.58 | 11.0 \| 28.2 \| 37.0 \| 23.8 |
| **30. Life around alcohol** | **0.77 [0.71; 0.82]** | **0.61** | **0.81** | **0.51** | **0.53** | **14.5 \| 29.5 \| 38.3 \| 17.6** |
| 31. Plan around alcohol | 0.75 [0.69; 0.80] | 0.64 | 0.81 | 0.52 | 0.50 | 19.8 \| 27.3 \| 37.0 \| 15.9 |
| 32. Control of myself | 0.47 [0.36; 0.57] | 0.48 | 0.88 | 0.65 | 0.48 | 19.8 \| 32.2 \| 31.3 \| 16.7 |
| **Sleep** | | | 0.87 | 0.77 | | |
| 33. Good night's sleep | 0.77 [0.71; 0.82] | 0.77 | 0.78 | | 0.55 | 19.4 \| 20.7 \| 35.2 \| 24.7 |
| **34. Sleep enough** | **0.77 [0.71; 0.82]** | **0.75** | **0.77** | | **0.52** | **20.3 \| 26.0 \| 32.2 \| 21.6** |
| **AQoLS-Brief** | | | 0.80 | 0.37 | | |
| 04. Physical activities | 0.60 [0.51; 0.68] | 0.61 | 0.76 | 0.36 | 0.51 | 21.6 \| 27.3 \| 26.4 \| 24.7 |
| 11. Family | 0.64 [0.56; 0.71] | 0.59 | 0.75 | 0.35 | 0.53 | 16.7 \| 31.7 \| 28.2 \| 23.3 |
| 18. Financial difficulties | 0.37 [0.25; 0.48] | 0.35 | 0.80 | 0.42 | 0.30 | 44.9 \| 27.8 \| 20.7 \| 06.6 |
| 21. Wasting my life | 0.67 [0.59; 0.74] | 0.53 | 0.75 | 0.34 | 0.65 | 05.7 \| 25.6 \| 36.6 \| 32.2 |
| 23. Worried about my life | 0.60 [0.51; 0.68] | 0.42 | 0.77 | 0.36 | 0.64 | 04.4 \| 24.7 \| 44.5 \| 26.4 |

*(Continued)*

**Table 3.** (Continued)

| Item | ir-ρ[a] | Upper-lower index | Cronbach α[b] | Average inter-item correlation [c] | d[d] | Item distribution |
|------|---------|-------------------|---------------|-----------------------------------|------|-------------------|
| 30. Life around alcohol | 0.62 [0.53; 0.69] | 0.53 | 0.76 | 0.35 | 0.53 | 14.5 \| 29.5 \| 38.3 \| 17.6 |
| 34. Sleep enough | 0.29 [0.17; 0.40] | 0.36 | 0.82 | 0.44 | 0.52 | 20.3 \| 26.0 \| 32.2 \| 21.6 |

[a]Item-rest correlation (ir-ρ), i.e., correlation between the item and the dimension's score formed by all other items. Items in bold are those retained for the dimension in the final AQoLS-Brief respecting the a priori rules of choice.

[b]Cronbach α with all items of the dimension, followed by Cronbach α excluding the particular item of the dimension (called α-drop). The lowest the α-drop for an item is, the more important this item is for this dimension. When the dimension has only 2 items $x_i$ and $x_j$ then α-drop for item $x_i$ is computed as $cov(x_i, x_j)/var(x_j)$ [32]

[c]The average interitem correlations with all items of the dimension, followed by the average interitem correlations excluding the particular item of the dimension (called AIIC-drop). The lowest the AIIC-drop for an item is, the more important this item is for this dimension. It cannot be calculated for dimension with only two items.

[d]Item difficulty: average score of the item divided by its range.

The dimension "Control" was a tie between item 30 "My life has revolved around alcohol" and item 31 "I have planned my days around alcohol" *ex aequo* on rule 1. We kept item 30 because it did better on rule 2.

Two dimensions showed ties on the *a priori* rule 2.

The dimension "Negative emotions" was a tie between item 22 "I have worried about the effect alcohol has been having on my health" and item 23 "I have worried about alcohol causing problems in my life" *ex aequo* on rule 2. We kept item 23 because it did better on rule 1 and was more universal given the high inequality regarding alcohol-related physical damages, particularly hepatic complications with the same level of alcohol intoxication.

The dimension "Sleep" was a tie between item 33 "I have not had a good night's sleep" and item 34 "I have not been getting enough sleep" *ex aequo* on rule 2. We kept item 34 because it did better on rule 1.

Analyzing all 24000 possible combinations of the seven items identified a maximum possible Cronbach alpha of 0.83 for 14 versions. Two of the best alternative versions had five items in common with the final AQoLS-Brief, except for the dimensions "Living conditions" and "Sleep". For these two dimensions, items 18 and 34 were respectively in the two alternative versions replaced by item 17 "money spent on alcohol"/item 24 "Appetite", and item 33 "good night's sleep". As the improvement of the Cronbach coefficient was very low, we kept the version elicited with the *a priori* rules. Items 17 and 24 were also less global as a subjective impact of alcohol than item 18. The additional psychometrics on item difficulty, discrimination and internal consistency comforted the choice from the priori rules for all chosen items with a d in the optimal range 0.5–0.8, an upper-lower index >0.2 and an average inter-item correlation within or the closest to the optimal range 0.2–0.4. Only financial difficulty showed a d <0.5 but all other items of this dimension did, so the choice to retain this item was kept.

## Internal consistency and external validity

**Internal consistency.** Cronbach α for the AQoLS-Brief was 0.80 [0.76; 0.84]. Spearman correlation between AQoLS and AQoLS-Brief was 0.95. Fig 1 presents Standardized loadings estimates of the CFA of the AQoLS-Brief.

## External validity

Table 4 shows AQoLS and AQoLS-Brief external validity. As expected, we showed a moderate positive correlation between AQoLS-Total/ AQoLS-Brief and the number of DSM-5 AUD criteria (0.40), the sadness dimension of the ANPS and the dimensions hopelessness and anxiety sensitivity of the SURPS. We found a low correlation with the length since the onset of alcohol

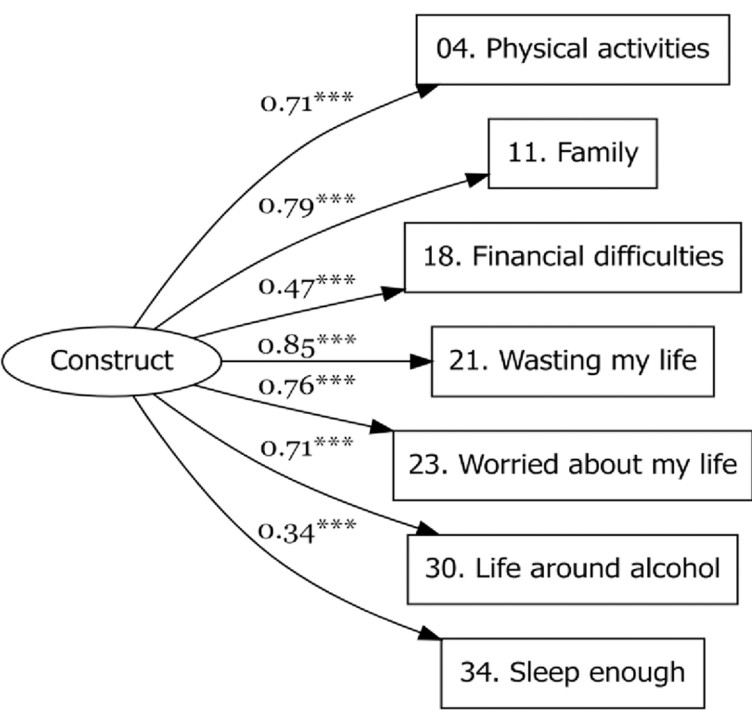

**Fig 1. Standardized loadings estimates of the confirmatory factor analysis using the diagonally weighted least squares method.**

*Legend: The overall model $\chi^2$ is 14.45 with 14 degrees of freedom. The p-value associated with this result is 0.417, which is non significant indicating that the observed covariance matrix matches the estimated covariance matrix within sampling variance. The value for root mean square error, an absolute fit index, is 0.012, i.e.,below the 0.08 guideline. The 90 percent confidence interval for this root mean square error is [0.000; 0.066]. Therefore, even the upper bound of root mean square error is below the 0.08 guideline. We conclude that the root mean square error provides additional support for model fit. The standardized root mean square residual with a value of 0.045 is below the 0.08 guideline. Moving to the incremental fit indices, the Tucker-Lewis Index and Comparative Fit Index indices have both the value 1.000 which is greater than the guideline 0.94. In conclusion, the CFA results suggest that the AQoLS-Brief measurement model provides a good fit.*

use disorder (0.22), with the number of HDD in the last 4-week drinking period (0.10) and the fear dimension of the ANPS. Similarly, we found no correlation with the MoCA score (-0.03). More surprisingly, we found no correlations with CGI-severity (0.03). Correlations with all variables and AQoLS were superimposable with those with AQoLS-Brief.

Table 5 provides a multiple linear regression between AQoLS-brief and external variables. Severity of alcohol use disorder, as measured by the number of DSM-5 alcohol criteria and the number ofHDD and the Dimension Sadness (ANPS) were the heaviest contributors to the AQoLS-Brief score. A younger age and length since the onset of alcohol use disorder also contributed significantly to the AQoLS -Brief score.

Table 4. Spearman correlations between external variables and AQoLS/AQoLS-Brief (coef [IC95%]).

| | AQoLS-Total | AQoLS-Brief |
|---|---|---|
| Age (years) | −0.16 [−0.29; −0.03] | −0.19 [−0.32; −0.06] |
| CGI-severity | 0.03 [−0.11; 0.17] | 0.04 [−0.09; 0.18] |
| Moca Score | −0.03 [−0.16; 0.11] | −0.06 [−0.20; 0.07] |
| Number of DSM-5 alcohol criteria | 0.40 [0.28; 0.51] | 0.40 [0.28; 0.51] |
| Length since the onset of alcohol use disorder (years) | 0.22 [0.08; 0.34] | 0.20 [0.07; 0.32] |
| Number of heavy drinking days (28d) | 0.10 [−0.04; 0.23] | 0.08 [−0.05; 0.22] |
| ANPS score | | |
| Fear | 0.19 [0.06; 0.32] | 0.16 [0.03; 0.29] |
| Sadness | 0.41 [0.29; 0.51] | 0.36 [0.23; 0.47] |
| SURPS score | | |
| Anxiety Sensitivity | 0.28 [0.16; 0.40] | 0.26 [0.13; 0.38] |
| Hopelessness | 0.32 [0.19; 0.43] | 0.29 [0.16; 0.41] |

Table 5. Multiple linear regression between AQoLS-Total/AQoLS-Brief and external variables.

| | AQoLS-Total Coef [CI95%] | AQoLS-Brief Coef [CI95%] | Variance Inflation Factor[a] [CI95%] |
|---|---|---|---|
| (Intercept) | −0.26** [−0.51; −0.02] | −0.27** [−0.52; −0.02] | |
| Age (years) | −0.21*** [−0.33; −0.10] | −0.24*** [−0.35; −0.12] | 1.16 [1.06; 1.44] |
| Number of DSM alcohol criteria | 0.33*** [0.22; 0.44] | 0.33*** [0.22; 0.45] | 1.12 [1.03; 1.43] |
| Length since the onset of alcohol use disorder (years) | 0.22*** [0.11; 0.33] | 0.21*** [0.09; 0.32] | 1.14 [1.05; 1.43] |
| Dimension Fear (ANPS) | −0.02 [−0.14; 0.11] | −0.03 [−0.16; 0.09] | 1.36 [1.20; 1.65] |
| Dimension Sadness (ANPS) | 0.31*** [0.17; 0.45] | 0.27*** [0.13; 0.41] | 1.72 [1.47; 2.09] |
| Dimension Anxiety Sensitivity (SURPS) | 0.05 [−0.07; 0.17] | 0.07 [−0.05; 0.19] | 1.29 [1.15; 1.57] |
| Dimension Hopelessness (SURPS) | 0.09 [−0.03; 0.22] | 0.09 [−0.04; 0.21] | 1.42 [1.24; 1.72] |
| **Number of heavy drinking days** | | | 1.08 [1.01; 1.48] |
| 1–12 days | Ref | Ref | |
| 13–27 days | 0.49*** [0.15; 0.84] | 0.50*** [0.15; 0.86] | |
| 28 days | 0.28* [−0.01; 0.56] | 0.28* [−0.01; 0.56] | |
| N | 224 | 224 | |
| R2 | 0.40 | 0.38 | |

a: check for multicollinearity among independant variables. Should be less than 5

Levels of statistical significance: * p<0.10, ** p<0.05, *** p<0.01

Dependant variables and continuous independant variables are standardized to facilitate comparison of estimates

## Discussion

The AQoLS-Brief, a 7-item short version of the AQoLS, has demonstrated good internal consistency and unidimensionality, supporting the calculation of a total score. It is a valid and user-friendly measure of the impact of alcohol on quality of life, as perceived by individuals with lived experience. By offering a more concise alternative, the AQoLS-Brief could alleviate barriers associated with the use of the longer AQoLS version, making it more practical for clinical and research settings. Furthermore, it meets the necessary standards for use in a future discrete choice experiment to develop a specific health utility score.. We found a moderate correlation with DSM-5 criteria for AUD, the sadness dimension of the ANPS and the

dimensions hopelessness and anxiety sensitivity of the SURPS, and low correlation with the length of alcohol use disorder and the number of HDD during the last 4-week drinking period and fear dimension of the ANPS, as expected. No correlation was found with the MoCA score and the CGI-Severity; CGI-severity was very low and probably underestimated by clinicians in our sample in which patients were recently detoxified and for some still in a controlled environment.

Our findings are consistent with the previous AQoLS validation study. Most of the items chosen were those with the heaviest load for their respective dimension on the exploratory factorial analysis from the initial validation sample [3], except for item 21 "wasting my life" and item 11 "family". However, item 21 appeared to be the item least sensitive to the image perceived by others, instead illustrating the self-image independent of the environment or close relatives, which then more accurately reflects the self-esteem dimension, a feeling towards oneself. Item 11 seemed less sensitive to living style and preferences than the other items of the "Relationships" dimension. Thus, regardless of living style, the subjective quality of family relations can be assessed according to personal values or standards. Moreover, these items refer to global life-values according to patients' own interpretation. This makes them particularly interesting when assessing emerging third wave therapies and mutual support, especially for holistic approaches (e.g., Acceptance and Commitment therapy, 12-step programs) on the difference from drinking outcomes, that emphasize life-values [33].

We confirmed our hypothesis of a moderate correlation between the AQoLS/AQoLS-brief and the related but different construct of clinician-assessed alcohol use disorder severity using the number of DSM-5 criteria. The moderate to low correlations between anxiety- and depression-related dimensions of the ANPS and SURPS were also expected. This aligns with findings that anxiety- and depression-related items in non-specific HRQOL instruments are often the only dimensions affected by alcohol, whereas other dimensions remain unaffected [13]. Consequently, such non-specific instruments fail to capture the full spectrum of alcohol's impact on HRQOL, supporting the use of specific tools, such as the AQoLS, which are designed to comprehensively assess the unique impact of alcohol on quality of life. The low correlation with the number of HDD during the last 4-week drinking period confirms the limitations of drinking outcomes limited to drinking intensity. We replicate here previous findings on the construct validity of the AQoLS [11], and the added value of measuring the impact of alcohol with the perspective of people with lived experience. The 7-day recall period of the CGI-Severity, assessed while patients were hospitalized in a controlled, alcohol-fee environment, likely underestimated the severity of alcohol use disorder and may have artificially weakened its association with the AQoLS-Brief. The very low level of the CGI-S with a 7-day recall period is a good illustration of the difficulty in assessing inpatients with alcohol use disorder, for whom the control of the stimuli could be a relief with a temporary suspension of symptoms associated with alcohol intoxication, particularly craving, and does not reflect the wide range of effects of alcohol on quality of life, as measured by the AQoLS and AQoLS-Brief. The lack of correlation between the AQoLS/ AQoLS-Brief and the MoCA shows that these measures are not sensitive to cognitive impairment, which is very common in alcohol use disorders [34]. The AQoLS and the AQoLS-Brief have the same results for internal and external validity, supporting the validity of the AQoLS-Brief.

Our study has certain limitations, as we chose not to conduct a new exploratory factor analysis, focusing instead on a confirmatory approach. This decision was made to retain the previously described seven-dimension structure, which reflects patients' perspectives, by selecting the best item from each dimension. The a priori rules for selecting these items were a posteriori validated by the additional analyses, which demonstrated the relevance of the selected items and the strong psychometric properties of the 7-item

AQoLS-Brief scale. We included a limited number of external variables to assess divergent and convergent validity. Notably, we did not measure quality of life using another instrument, as the data were collected as part of a randomized clinical trial, and adding another measure would have been too time-consuming. Our population is a sample of people with severe AUD, and is therefore not representative of all people with AUD, included in a randomized controlled trial, particularly light or mild AUD. However the whole range of AQoLS scores was covered by our sample, which allowed to perform analyses with no floor effect.

## Conclusion

Our study supports the aim of developing a concise and practical measure of quality of life in individuals with AUD through the creation of the AQoLS-Brief. We successfully validated this short version, which retains the essential dimensions of the original AQoLS. Moreover, our findings underscore the relevance of patient-reported outcomes in evaluating quality of life, particularly in individuals recently detoxified, where clinician-rated assessments may fall short. By providing a concise yet comprehensive assessment of quality of life, this instrument has the potential to enhance our understanding of patient preferences and priorities, guiding the delivery of more effective and tailored interventions in both clinical and research settings. The AQoLS-Brief appears to be a valid and easy to use measure of the impact of alcohol on quality of life from the perspective of people with lived experience. It meets the standards for a future discrete choice experiment study to develop a specific health utility score. This short version is then a significant step toward establishing a relevant and specific preference-based measure for calculating QALYs in AUD populations.

## Author contributions

**Conceptualization:** Amandine Luquiens, Arnaud Carré, Nathalie Pelletier-Fleury.

**Data curation:** Henri Panjo.

**Formal analysis:** Amandine Luquiens, Henri Panjo.

**Funding acquisition:** Amandine Luquiens.

**Investigation:** Amandine Luquiens, Alexandra Dereux, Patrice Louville, Hélène Donnadieu, Marie Bronnec, Amine Benyamina, Pascal Perney.

**Methodology:** Amandine Luquiens, Henri Panjo, Arnaud Carré, Nathalie Pelletier-Fleury.

**Project administration:** Amandine Luquiens.

**Software:** Henri Panjo.

**Supervision:** Amandine Luquiens, Nathalie Pelletier-Fleury.

**Validation:** Amandine Luquiens, Henri Panjo, Nathalie Pelletier-Fleury.

**Writing – original draft:** Amandine Luquiens.

**Writing – review & editing:** Henri Panjo, Alexandra Dereux, Patrice Louville, Hélène Donnadieu, Marie Bronnec, Amine Benyamina, Pascal Perney, Arnaud Carré, Nathalie Pelletier-Fleury.

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
