## [Decision Letter · Decision Letter 0]

3 Jan 2025

PONE-D-24-44639AQoLS-Brief: development and psychometric properties of a short version of the Alcohol Quality of Life ScalePLOS ONE

Dear Dr. Luquiens,

Thank you for submitting your manuscript to PLOS ONE. After careful consideration, we feel that it has merit but does not fully meet PLOS ONE’s publication criteria as it currently stands. Therefore, we invite you to submit a revised version of the manuscript that addresses the points raised during the review process.

We look forward to receiving your revised manuscript.

Kind regards,

Shivanand Kattimani

Academic Editor

PLOS ONE

“The TRAIN study was funded by a French national grant of the ministry of health PHRC-N 2016.”

“AL, HP, MB, AD, PP, AC, NPF: no conflict of interest related to this manuscript

HD: declares to have participated in occasional interventions (conference activities) for Abbvie, Gilead, Ethypharm and consultancy for Ethypharm

AB: declares to have participated in occasional interventions (conference activities) for Eisai, Gilead and Abbvie. Board member Indivior and Camurus

PL: declares to have participated in occasional interventions (conference activities) for Lundbeck.”

Additional Editor Comments:

kindly see the reviewers comments and address their queries.

We look forward to your revision within four weeks of time.

If you want you can request extension.

Reviewers' comments:

Reviewer's Responses to Questions

**Comments to the Author**

1. Is the manuscript technically sound, and do the data support the conclusions?

Reviewer #1: Yes

Reviewer #2: Partly

2. Has the statistical analysis been performed appropriately and rigorously? 

Reviewer #1: Yes

Reviewer #2: Yes

3. Have the authors made all data underlying the findings in their manuscript fully available?

Reviewer #1: No

Reviewer #2: Yes

4. Is the manuscript presented in an intelligible fashion and written in standard English?

Reviewer #1: No

Reviewer #2: Yes

5. Review Comments to the Author

Reviewer #1: Introduction

1. Line 43: Add a concise explanation of the COSMIN criteria and highlight their relevance.

2. Line 43: While the Alcohol Quality of Life Scale (AQoLS) is described as having excellent internal consistency, other psychometric properties (e.g., test-retest reliability, validity in different populations) of both the original and brief versions are not addressed. Adding this would make the justification for the AQoLS more robust.

3. Line 54: ‘A shorter version would be more convenient for both researchers and users.’ -- What was the rationale for using the shorter version of AQoLS-Brief? It is not clear. Clearly state the research gap this study aims to fill.

4. Provide more specifics on how the AQoLS-Brief will facilitate discrete choice experiments. Will it reduce complexity or address specific methodological challenges? This clarification will underline the importance of this study.

Study Design and Methods

1. Line 81: Write the full abbreviation of TRAIN study.

2. Line 97: ‘Patient had to be aged 18 to 65 with current AUD, according to the DSM-5, classed at least as high drinking risk level (men: alcohol consumption >60 g/day; women >40 g/day) as regarding the alcohol consumption in the last 4-week drinking period.’ -- break it into two lines.

3. Line 100: Why these specific criteria (e.g., 7-30 days abstinence) were selected. Does this enhance the reliability of baseline data, reduce variability, or align with clinical practice?

4. Please note that the inclusion criteria started from lines 97 – 103. Line 100: include ‘also’ – Inclusion criteria also included no benzodiazepines use for at least 3 days prior to inclusion, to avoid interference with alcohol intoxication or withdrawal medication on the neuropsychological assessments, and any recall bias on drinking.

5. Consider adding more discussion on how the AQoLS-Brief was developed with input from patients and how this ensures its relevance.

6. Line 171: Write the full abbreviation of MoCA score.

Discussion

The discussion could better connect the results to the study’s objectives and implications for future research or clinical practice. Discuss how the AQoLS-Brief improves on the original AQoLS in terms of usability and application. Address potential limitations of the study (e.g., sample representativeness, reliance on specific statistical rules for item selection) and suggest how they could be addressed in future research.

Reviewer #2: I have also attached a document with the detailed reviews.

Thank you for the opportunity to review this manuscript. The current study creates and validates a brief version of the Alcohol Quality of Life Scale (AQoLS-Brief) in a sample of recently detoxified patients with alcohol use disorder. Please find me critiques below.

• The external validity variables, besides the MoCA, were very focused on alcohol use-related constructs. The variables included are good, but some additional variables not related to alcohol specifically but that map onto some of the subscales or global quality of life would strengthen the convergent and discriminant validity analysis (Clark & Watson, 2019). Candidate variables could include scales of global quality of life, depression, anxiety, physical health pr medical history, relationship functioning, socioeconomic status, sleep quality, and so on. If none of these variables were measured in the original data collection, then at the very least the lack of these measures should be included as a limitation.

o Clark, L. A., & Watson, D. (2019). Constructing validity: New developments in creating objective measuring instruments. Psychological Assessment, 31(12), 1412–1427. https://doi.org/10.1037/pas0000626

• Related to the point above, it would be useful to examine validity predictors in a multiple regression together (as opposed to bivariate correlations only). This would allow for parsing apart which constructs are most strongly/uniquely associated with the AQoLS-Brief.

• I am confused why it was hypothesized that there would be no correlation between the AQoLS-Brief and cognitive impairment. Cognitive impairment is a well-established correlate of alcohol use disorder (Bernardin et al., 2014; Wang et al., 2023), even after recovery (Stavro et al., 2013). The current manuscript even cited an article demonstrating the utility of the MoCA as a screening tool for alcohol-dependent individuals, and the data comes from a clinical trial in which patients undergo a cognitive training program. There is also evidence that cognitive impairment and quality of life are negatively associated (Hill et al., 2017). What is the rationale that alcohol-related impairment in life would not be associated with cognitive impairment?

o Bernardin, F., Maheut-Bosser, A., & Paille, F. (2014). Cognitive impairments in alcohol-dependent subjects. Frontiers in psychiatry, 5, 78.

o Hill, N. L., McDermott, C., Mogle, J., Munoz, E., DePasquale, N., Wion, R., & Whitaker, E. (2017). Subjective cognitive impairment and quality of life: a systematic review. International psychogeriatrics, 29(12), 1965-1977.

o Stavro, K., Pelletier, J., & Potvin, S. (2013). Widespread and sustained cognitive deficits in alcoholism: a meta‐analysis. Addiction biology, 18(2), 203-213.

o Wang, G., Li, D. Y., Vance, D. E., & Li, W. (2023). Alcohol use disorder as a risk factor for cognitive impairment. Journal of Alzheimer's Disease, 94(3), 899-907.

• I have some reservations about the decision to exclude factorial analysis and believe that a confirmatory factor analysis (CFA) should be estimated. I understand trying to retain the original 7-dimension structure, but surely the intention of this scale is to be used as a screen for general alcohol-related quality of life impairment, suggesting the measure will be treated as a single factor scale. A CFA would inform how the measure performs as a single, unidimensional factor of alcohol-related quality of life impairment (Briggs & Cheek, 1986; Simms, 2008). Examining factor loading strength and patterns of the brief dimension could even be used to guide item selection. Currently as it stands, the two selection criteria tell us whether the items selected are representative of their original scale? A CFA would tell us do these selected items hang together as a single construct that can be used meaningfully? Moreover, the original studies examining psychometric performance of the AQoLS reported: “The preliminary principal component analysis indicated a substantive principal dimension accounting for 42 % of the variance, thus indicating the appropriateness of a total AQoLS score summing the items” (Luquiens et al., 2016). While a 7-factor solution was ultimately settled on, it seems that testing a single factor CFA is appropriate given some of the preliminary work with the full measure and the purpose of using brief measures more broadly.

o Briggs SR, & Cheek JM (1986). The role of factor analysis in the development and evaluation of personality scales. Journal of Personality, 54, 106–148.> doi: 10.1111/j.1467-6494.1986.tb00391

o Luquiens, A., Whalley, D., Laramée, P., Falissard, B., Kostogianni, N., Rehm, J., ... & Aubin, H. J. (2016). Validation of a new patient-reported outcome instrument of health-related quality of life specific to patients with alcohol use disorder: the Alcohol Quality of Life Scale (AQoLS). Quality of Life Research, 25, 1549-1560.

o Simms, L. J. (2008). Classical and Modern Methods of Psychological Scale Construction. Social and Personality Psychology Compass, 2(1), 414–433. https://doi.org/10.1111/j.1751-9004.2007.00044.x

• I believe the effect of the sample and assessment time periods on validity associations needs to be discussed more in the manuscript. For example, in the discussion, it is stated that the low correlation between the number of HDD in the last 4 weeks and the AQoLS-brief suggests that drinking intensity has limitations as an outcome. While there is some truth to this statement, I think the association between the number of HDD in the last 4 weeks and the AQoLS-brief needs some more discussion. If I am understanding the inclusion criteria correctly, patients recently detoxified and, at minimum, did not drink in the last 7 days (up to 30). This means that heavy drinking for individuals was at minimum ¼ lower, and potentially completely absent, in the assessment period. Consequently, this inclusion criteria artificially attenuates the association between drinking intense and quality of life. In my opinion, and if I am understanding things correctly, this is likely the best explanation for why heavy drinking and quality of life from drinking were not significantly associated with one another. The conclusion above about this association should be tempered.

o The association with the CGI-severity score has a similar issue. It is concluded that assessing impairment in an inpatient setting does not capture the full spectrum of quality-of-life alcohol impairment. Again, I do not inherently disagree with this statement, but the past 7 days, by definition of the inclusion criteria, means that there was not alcohol use present during the time of the CGI-severity assessment. Would this not artificially decrease the disease severity score to be lower and thus attenuate the CGI-severity and AQoLS-brief association?

• I would like to hear more information about excluding patients currently experiencing withdrawal symptoms from the current analysis. What proportion of the sample does this exclude? Does this only include extreme, life-threatening withdrawal symptoms, or even more mild withdrawal symptoms? If it is life-threatening symptoms only then that makes sense for safety purposes. However, exclusion based on mild withdrawal symptoms may bias the sample unnecessarily.

• I am wondering why other item selection analyses were not used to guide item selection, such as item discrimination and item difficulty. Some rationale as to why these were not used would be helpful.

• This is a minor, but there are so many acronyms it is difficult to follow the manuscript at times. I recommend only using very common acronyms (e.g., AUD, DSM-5) and writing out most other acronyms.

• This is also minor, but please proofread the manuscript for correct grammar. For example, on lines 52-54, it reads: “However, its length could limit its use in clinical trials, where subjective measurements are not often the primary judgment outcome (Luquiens et al. 2012) and there is often numerous and time spending measures.” Instead of “there is often numerous and time spending measures,” I assume the authors mean, “there is often numerous and time intensive measures.” There are several places throughout the manuscript where the wording or grammar is slightly off.

6. PLOS authors have the option to publish the peer review history of their article (what does this mean? ). If published, this will include your full peer review and any attached files.

**Do you want your identity to be public for this peer review?** For information about this choice, including consent withdrawal, please see our Privacy Policy .

Reviewer #1: No

Reviewer #2: No

---

## [Author Response · Author response to Decision Letter 1]

28 Jan 2025

We would like to thank the reviewers for their thoughtful comments and suggestions, which have helped improve the quality of our manuscript. Below, we provide our detailed responses to each of the comments, outlining the revisions made and clarifications provided to address the points raised.

Reviewer #1: Introduction

1. Line 43: Add a concise explanation of the COSMIN criteria and highlight their relevance.

R1. We reformulated the 1st paragraph to explain the COSMIN criteria and highlight their relevance: P4L41“The Alcohol Quality of Life Scale (AQoLS) is a patient-reported outcome measure (PROM) developed with significant input from patients with current or remitted alcohol use disorder (AUD) (Luquiens et al. 2015), in accordance with COSMIN (COnsensus-based Standards for the selection of health Measurement INstruments) guidelines (Mokkink et al. 2018). These guidelines emphasize ensuring strong content validity by involving patients in identifying relevant elements through concept elicitation and conducting cognitive interviews to assess the comprehensiveness, comprehensibility, and relevance of the items in alignment with the intended construct”

2. Line 43: While the Alcohol Quality of Life Scale (AQoLS) is described as having excellent internal consistency, other psychometric properties (e.g., test-retest reliability, validity in different populations) of both the original and brief versions are not addressed. Adding this would make the justification for the AQoLS more robust.

R2. We added the following sentence to provide additional information on the psychometrics of the AQoLS: P4L589“The AQoLS total score showed an intraclass correlation coefficient at 2 weeks in stable out-patients with alcohol use disorder of 0.80 (Higuchi, Moriguchi, and Tan 2020). “As the Brief version was not developed, we cannot provide further psychometrics of the AQoLS-Brief in other populations.

3. Line 54: ‘A shorter version would be more convenient for both researchers and users.’ -- What was the rationale for using the shorter version of AQoLS-Brief? It is not clear. Clearly state the research gap this study aims to fill.

R3. We added the following explanation in the introduction to clarify the rationale for using a shorter version of the AQoLS:: P4L63“Using a shorter version of a scale is more convenient as it reduces the time burden on both participants and researchers, making it easier to administer in clinical or research settings where time constraints are often present. A shorter scale also improves participant engagement and completion rates, while still retaining the key measurement properties and relevance of the longer version. This helps ensure more efficient data collection without compromising the quality of the information gathered.“

4. Provide more specifics on how the AQoLS-Brief will facilitate discrete choice experiments. Will it reduce complexity or address specific methodological challenges? This clarification will underline the importance of this study.

R4. We provided more details: P5L87 “Short versions of scales are needed for a Discrete Choice Experiment due to the complexity of constructing pairwise combinations. These combinations involve comparing multiple health states defined by different attributes, and as the number of attributes and levels increases, the total number of possible pairwise combinations grows exponentially. By using condensed versions of scales, which reduces the number of attributes and levels, the number of combinations is significantly reduced, making the experimental design more manageable. This simplification helps avoid participant fatigue and ensures the choice tasks are feasible and understandable, leading to more reliable data for deriving utility values.”

Study Design and Methods

5. Line 81: Write the full abbreviation of TRAIN study.

R5. We detailed P6L110“TRAIN study (TRAining INhibition in alcohol use disorder)”

6. Line 97: ‘Patient had to be aged 18 to 65 with current AUD, according to the DSM-5, classed at least as high drinking risk level (men: alcohol consumption >60 g/day; women >40 g/day) as regarding the alcohol consumption in the last 4-week drinking period.’ -- break it into two lines.

R6. We broke it into two lines.

7. Line 100: Why these specific criteria (e.g., 7-30 days abstinence) were selected. Does this enhance the reliability of baseline data, reduce variability, or align with clinical practice?

R7. These criteria were necessary as the intervention aimed to prevent relapse in individuals with alcohol use disorder who had recently undergone detoxification. They were also required to enhance the reliability of baseline data and to minimize recall bias related to describing the last drinking period.

8. Please note that the inclusion criteria started from lines 97 – 103. Line 100: include ‘also’ – Inclusion criteria also included no benzodiazepines use for at least 3 days prior to inclusion, to avoid interference with alcohol intoxication or withdrawal medication on the neuropsychological assessments, and any recall bias on drinking.

R8. We included “also”.

9. Consider adding more discussion on how the AQoLS-Brief was developed with input from patients and how this ensures its relevance.

R9. We modified a couple of paragraphs in the introduction and methods: P4L41“The Alcohol Quality of Life Scale (AQoLS) is a patient-reported outcome measure (PROM) developed with significant input from patients with current or remitted alcohol use disorder (AUD)”. P8L193 “We selected one item per dimension using a priori rules demonstrating the predominant nature of the item within each dimension. This choice was made to respect the input of patients at the time of the development of the AQoLS, that lead in a qualitative analysis to the identification of the 7 dimensions (Luquiens et al. 2015) confirmed in the later validation study (Luquiens et al. 2016).”

10. Line 171: Write the full abbreviation of MoCA score.

R10. We corrected for: “Montreal Cognitive Assessment (MoCA)”

Discussion

11. The discussion could better connect the results to the study’s objectives and implications for future research or clinical practice. Discuss how the AQoLS-Brief improves on the original AQoLS in terms of usability and application. Address potential limitations of the study (e.g., sample representativeness, reliance on specific statistical rules for item selection) and suggest how they could be addressed in future research.

R11. We modified the discussion: P19L339 “The AQoLS-Brief, a 7-item short version of the AQoLS, has demonstrated good internal consistency and unidimensionality, supporting the calculation of a total score. It is a valid and user-friendly measure of the impact of alcohol on quality of life, as perceived by individuals with lived experience. By offering a more concise alternative, the AQoLS-Brief could alleviate barriers associated with the use of the longer AQoLS version, making it more practical for clinical and research settings. Furthermore, it meets the necessary standards for use in a future discrete choice experiment to develop a specific health utility score.”

We added several limits in the discussion: P20L387“Our study has certain limitations, as we chose not to conduct a new exploratory factor analysis, focusing instead on a confirmatory approach. This decision was made to retain the previously described seven-dimension structure, which reflects patients’ perspectives, by selecting the best item from each dimension. The a priori rules for selecting these items were a posteriori validated by the additional analyses, which demonstrated the relevance of the selected items and the strong psychometric properties of the 7-item AQoLS-Brief scale. We included a limited number of external variables to assess divergent and convergent validity. Notably, we did not measure quality of life using another instrument, as the data were collected as part of a randomized clinical trial, and adding another measure would have been too time-consuming.”

Reviewer #2:

The external validity variables, besides the MoCA, were very focused on alcohol use-related constructs. The variables included are good, but some additional variables not related to alcohol specifically but that map onto some of the subscales or global quality of life would strengthen the convergent and discriminant validity analysis (Clark & Watson, 2019). Candidate variables could include scales of global quality of life, depression, anxiety, physical health pr medical history, relationship functioning, socioeconomic status, sleep quality, and so on. If none of these variables were measured in the original data collection, then at the very least the lack of these measures should be included as a limitation.

Clark, L. A., & Watson, D. (2019). Constructing validity: New developments in creating objective measuring instruments. Psychological Assessment, 31(12), 1412–1427. https://doi.org/10.1037/pas0000626

R1. We thank the reviewer for his/her request and suggestions for further analysis of external validity analyses.

We added psychological variables in the manuscript that were not reported before that can reflect anxiety and depression:

Methods: P8L181 “Psychological variables - We collected the Affective Neuroscience Personality Scales (ANPS) and the Substance Use Risk Profile Scale (SURPS) to assess personality traits. The ANPS measures six primary emotional traits: seeking, care, play, fear, anger, and sadness. It was initially validated by Davis and Panksepp (Davis, Panksepp, and Normansell 2003) providing a framework to explore the neurobiological basis of personality. The SURPS assesses four personality dimensions associated with substance use vulnerability: anxiety sensitivity, hopelessness, impulsivity, and sensation seeking, with its initial validation conducted by Woicik et al. (Woicik et al. 2009). Both scales are psychometrically robust and widely used in research to link personality traits to behavioral and emotional outcomes.“

P10L236 “We hypothesized a moderate correlation […] with dimensions of the psychological scales ANPS and SURPS related to anxiety-depression, i.e. ANPS sadness and fear dimensions, and SURPS anxiety sensitivity and hopelessness dimensions.”

P16L319 AND 323 “Results:

“We found a moderate positive correlation […] with sadness dimension of the ANPS and the dimensions hopelessness and anxiety sensitivity of the SURPS.”

AQoLS-Total AQoLS-Brief

Age (years) -0.16 [-0.29; -0.03] -0.19 [-0.32; -0.06]

CGI-severity 0.03 [-0.11; 0.17] 0.04 [-0.09; 0.18]

Moca Score -0.03 [-0.16; 0.11] -0.06 [-0.20; 0.07]

Number of DSM alcohol criteria 0.40 [0.28; 0.51] 0.40 [0.28; 0.51]

Length since the onset of alcohol use disorder (years) 0.22 [0.08; 0.34] 0.20 [0.07; 0.32]

Number of heavy drinking days (28d) 0.10 [-0.04; 0.23] 0.08 [-0.05; 0.22]

ANPS score

Fear 0.19 [0.06; 0.32] 0.16 [0.03; 0.29]

Sadness 0.41 [0.29; 0.51] 0.36 [0.23; 0.47]

SURPS score

Anxiety Sensitivity 0.28 [0.16; 0.40] 0.26 [0.13; 0.38]

Hopelessness 0.32 [0.19; 0.43] 0.29 [0.16; 0.41]

Discussion: P19L359 and 366 “We found a moderate correlation with DSM-5 criteria for AUD, the sadness dimension of the ANPS and the dimensions hopelessness and anxiety sensitivity of the SURPS. ““The moderate to low correlations between anxiety- and depression-related dimensions of the ANPS and SURPS were also expected. This aligns with findings that anxiety- and depression-related items in non-specific HRQOL instruments are often the only dimensions affected by alcohol, whereas other dimensions remain unaffected (Luquiens et al. 2012). Consequently, such non-specific instruments fail to capture the full spectrum of alcohol's impact on HRQOL, supporting the use of specific tools, such as the AQoLS, which are designed to comprehensively assess the unique impact of alcohol on quality of life”

We also added the description of the 2 scales in table 1. P11

Variable N=227

ANPS total score (mean, sd) 63.3 (9.7)

Sadness 10.2 (3.3)

Fear 10.1 (3.7)

Care 12.8 (3.1)

Anger 7.4 (3.5)

Play 10.9 (3.3)

Seeking 11.9 (3.1)

SURPS Total score (mean, sd) 56.1 (8.0)

Hopelessness 16.7 (4.3)

Anxiety Sensitivity 15.2 (2.9)

Sensation Seeking 13.0 (3.4)

Impulsivity 11.4 (3.0)

Moreover, we added in the limit section the following limit: P21L400 “We included a limited number of external variables to assess divergent and convergent validity. Notably, we did not measure quality of life using another instrument, as the data were collected as part of a randomized clinical trial, and adding another measure would have been too time-consuming.”

Related to the point above, it would be useful to examine validity predictors in a multiple regression together (as opposed to bivariate correlations only). This would allow for parsing apart which constructs are most strongly/uniquely associated with the AQoLS-Brief.

R2. As suggested by the reviewer, we provided a multiple linear regression between AQoLS-brief and the external variables:

Methods:

P10L243” We performed a multiple linear regression explaining the AQoLS-brief/AQoLS by the external variables.”

Results: P18L328 “Table 6 provides a multiple linear regression between AQoLS-brief and external variables. Severity of alcohol use disorder, as measured by the number of DSM-5 alcohol criteria, the HDD and the Dimension Sadness (ANPS) were the heaviest contributors to the AQoLS-Brief score. A younger age and length since the onset of alcohol use disorder also contributed significantly to the AQoLS -Brief score. “

Table 6: Multiple linear regression between AQoLS-Total /AQoLS-Brief and external variables

AQoLS-Total

Coef [CI95%] AQoLS-Brief

Coef [CI95%] Variance Inflation Factora [CI95%]

(Intercept) -0.26** [-0.51; -0.02] -0.27** [-0.52; -0.02]

Age (years) -0.21*** [-0.33; -0.10] -0.24*** [-0.35; -0.12] 1.16 [1.06; 1.44]

Number of DSM alcohol criteria 0.33*** [0.22; 0.44] 0.33*** [0.22; 0.45] 1.12 [1.03; 1.43]

Length since the onset of alcohol use disorder (years) 0.22*** [0.11; 0.33] 0.21*** [0.09; 0.32] 1.14 [1.05; 1.43]

Dimension Fear (ANPS) -0.02 [-0.14; 0.11] -0.03 [-0.16; 0.09] 1.36 [1.20; 1.65]

Dimension Sadness (ANPS) 0.31*** [0.17; 0.45] 0.27*** [0.13; 0.41] 1.72 [1.47; 2.09]

Dimension Anxiety Sensitivity (SURPS) 0.05 [-0.07; 0.17] 0.07 [-0.05; 0.19] 1.29 [1.15; 1.57]

Dimension Hopelessness (SURPS) 0.09 [-0.03; 0.22] 0.09 [-0.04; 0.21] 1.42 [1.24; 1.72]

Number of heavy drinking days 1.08 [1.01; 1.48]

1-12 days Ref Ref

13-27 days 0.49*** [0.15; 0.84] 0.50*** [0.15; 0.86]

28 days 0.28* [-0.01; 0.56] 0.28* [-0.01; 0.56]

N 224 224

R2 0.40 0.38

a: check for multicollinearity among independant variables. Should be less than 5

Levels of statistical significance: * p<0.10, ** p<0.05, *** p<0.01

Dependant variables and continuous independant variables are standardized to facilitate comparison of estimates

Methods were amended in accordance: P9L243 “We performed a multiple linear regression explaining the AQoLS-brief/AQoLS by the external variables.”

I am confused why it was hypothesized that there would be no correlation between the AQoLS-Brief and cognitive impairment. Cognitive impairment is a well-established correlate of alcohol use disorder (Bernardin et al., 2014; Wang et al., 2023), even after recovery (Stavro et al., 2013). The current manuscript even cited an article demonstrating the utility of the MoCA as a screening tool for alcohol-dependent individuals, and the data comes from a clinical trial in which patients undergo a cognitive training program. There is also evidence that cognitive impairment and quality of life are negatively associated (Hill et al., 2017). What is the rationale that alcohol-related impairment in life would not be associated with cognitive impairment?

Bernardin, F., Maheut-Bosser, A., & Paille, F. (2014). Cognitive impairments in alcohol-dependent subjects. Frontiers in psychiatry, 5, 78.

Hill, N. L., McDermott, C., Mogle, J., Munoz, E., DePasquale, N., Wion, R., & Whitaker, E. (2017). Subjective cognitive impairment and quality of life: a systematic review. International psychogeriatrics, 29(12), 1965-1977.

Stavro, K., Pelletier, J., & Potvin, S. (2013). Widespread and sustained cogni

---

## [Decision Letter · Decision Letter 1]

23 Feb 2025

PONE-D-24-44639R1AQoLS-Brief: development and psychometric properties of a short version of the Alcohol Quality of Life ScalePLOS ONE

Dear Dr. Luquiens,

Thank you for submitting your manuscript to PLOS ONE. After careful consideration, we feel that it has merit but does not fully meet PLOS ONE’s publication criteria as it currently stands. Therefore, we invite you to submit a revised version of the manuscript that addresses the points raised during the review process.

We look forward to receiving your revised manuscript.

Kind regards,

Shivanand Kattimani

Academic Editor

PLOS ONE

Journal Requirements:

Reviewers' comments:

Reviewer's Responses to Questions

**Comments to the Author**

1. If the authors have adequately addressed your comments raised in a previous round of review and you feel that this manuscript is now acceptable for publication, you may indicate that here to bypass the “Comments to the Author” section, enter your conflict of interest statement in the “Confidential to Editor” section, and submit your "Accept" recommendation.

Reviewer #1: All comments have been addressed

Reviewer #2: All comments have been addressed

2. Is the manuscript technically sound, and do the data support the conclusions?

Reviewer #1: Yes

Reviewer #2: Yes

3. Has the statistical analysis been performed appropriately and rigorously? 

Reviewer #1: Yes

Reviewer #2: Yes

4. Have the authors made all data underlying the findings in their manuscript fully available?

Reviewer #1: Yes

Reviewer #2: No

5. Is the manuscript presented in an intelligible fashion and written in standard English?

Reviewer #1: Yes

Reviewer #2: Yes

6. Review Comments to the Author

Reviewer #1: (No Response)

Reviewer #2: The authors have thoroughly responded to all my critiques. I commend the authors for their comprehensive and well-reasoned responses. The changes made have greatly strengthened the manuscript and increased the manuscripts contribution to the literature. I have one final minor comment to be addressed:

1. In the hypotheses (starting line 215), it would be helpful to specify the hypothesized directions of the associations (positive or negative) in addition to the strength of the correlations.

7. PLOS authors have the option to publish the peer review history of their article (what does this mean? ). If published, this will include your full peer review and any attached files.

**Do you want your identity to be public for this peer review?** For information about this choice, including consent withdrawal, please see our Privacy Policy .

Reviewer #1: No

Reviewer #2: No

---

## [Author Response · Author response to Decision Letter 2]

24 Feb 2025

We would like to thank the reviewers for their positive feedback. Below, we provide our detailed response to the last comment.

Reviewer #2: The authors have thoroughly responded to all my critiques. I commend the authors for their comprehensive and well-reasoned responses. The changes made have greatly strengthened the manuscript and increased the manuscripts contribution to the literature. I have one final minor comment to be addressed:

1. In the hypotheses (starting line 215), it would be helpful to specify the hypothesized directions of the associations (positive or negative) in addition to the strength of the correlations.

R1. We added that we expected positive correlations : P9L215: “We hypothesized a moderate positive correlation with measures of severity (CGI-severity and DSM-5 criteria for AUD), a moderate to low positive correlation with the history of alcohol use disorder (i.e the length since the onset of alcohol use disorder), the drinking outcome (i.e. the number of HDD in the last 4-week drinking period) and dimensions of the psychological scales ANPS and SURPS related to anxiety-depression, i.e. ANPS Sadness and Fear dimensions, and SURPS Anxiety Sensitivity and Hopelessness dimensions.”

---

## [Editor Report · Decision Letter 2]

5 Mar 2025

AQoLS-Brief: development and psychometric properties of a short version of the Alcohol Quality of Life Scale

PONE-D-24-44639R2

Dear Dr. Luquiens,

We’re pleased to inform you that your manuscript has been judged scientifically suitable for publication and will be formally accepted for publication once it meets all outstanding technical requirements.

Kind regards,

Shivanand Kattimani

Academic Editor

PLOS ONE
---

## [Editor Report · Acceptance letter]

PONE-D-24-44639R2

PLOS ONE

Dear Dr. Luquiens,

I'm pleased to inform you that your manuscript has been deemed suitable for publication in PLOS ONE. Congratulations! Your manuscript is now being handed over to our production team.

Kind regards,

on behalf of

Dr. Shivanand Kattimani

Academic Editor

PLOS ONE